# Analyzing the Interplay between COVID-19 Viral Load, Inflammatory Markers, and Lymphocyte Subpopulations on the Development of Long COVID

**DOI:** 10.3390/microorganisms11092241

**Published:** 2023-09-06

**Authors:** Andrea Rivera-Cavazos, José Antonio Luviano-García, Arnulfo Garza-Silva, Devany Paola Morales-Rodríguez, Mauricio Kuri-Ayache, Miguel Ángel Sanz-Sánchez, Juan Enrique Santos-Macías, Maria Elena Romero-Ibarguengoitia, Arnulfo González-Cantú

**Affiliations:** 1Vicerrectoría de Ciencias de la Salud, Escuela de Medicina, Universidad de Monterrey, San Pedro Garza García 66238, Nuevo León, Mexico; andrea.riverac@udem.edu (A.R.-C.);; 2Research Department, Hospital Clínica Nova de Monterrey, San Nicolás de los Garza 66450, Nuevo León, Mexico; drgzzcantu@gmail.com; 3Internal Medicine Department, Hospital Clínica Nova de Monterrey, San Nicolás de los Garza 66450, Nuevo León, Mexico; 4Cardiology Department, Hospital Clínica Nova de Monterrey, San Nicolás de los Garza 66450, Nuevo León, Mexico

**Keywords:** COVID-19, SARS-CoV-2 infection, long COVID, natural killer cells, lymphocytes

## Abstract

The global impact of the SARS-CoV-2 infection has been substantial, affecting millions of people. Long COVID, characterized by persistent or recurrent symptoms after acute infection, has been reported in over 40% of patients. Risk factors include age and female gender, and various mechanisms, including chronic inflammation and viral persistence, have been implicated in long COVID’s pathogenesis. However, there are scarce studies in which multiple inflammatory markers and viral load are analyzed simultaneously in acute infection to determine how they predict for long COVID at long-term follow-up. This study explores the association between long COVID and inflammatory markers, viral load, and lymphocyte subpopulation during acute infection in hospitalized patients to better understand the risk factors of this disease. This longitudinal retrospective study was conducted in patients hospitalized with COVID-19 in northern Mexico. Inflammatory parameters, viral load, and lymphocyte subpopulation during the acute infection phase were analyzed, and long COVID symptoms were followed up depending on severity and persistence (weekly or monthly) and assessed 1.5 years after the acute infection. This study analyzed 79 patients, among them, 41.8% presented long COVID symptoms, with fatigue being the most common (45.5%). Patients with long COVID had higher lymphocyte levels during hospitalization, and NK cell subpopulation levels were also associated with long COVID. ICU admission during acute COVID-19 was also linked to the development of long COVID symptoms.

## 1. Introduction

The infection caused by SARS-CoV-2, which originated in Wuhan, China, in December 2019, has had a significant impact globally. To date, it is estimated to have infected more than 767 million people and has been responsible for nearly 7 million deaths [1].

The term “Long-COVID” is used to describe the presence of prolonged or recurrent symptoms that persist for at least four weeks after an acute infection with the SARS-CoV-2 virus and that cannot be attributed to any other disease, according to the US Centers for Disease Control and Prevention (CDC) [2,3,4,5]. Long COVID has been reported to affect more than 40% of patients who experienced acute COVID-19 infection, and there are even investigations that found its presence in up to 60% of patients [6,7]. The associated symptoms are diverse, although fatigue is one of the most frequent, and all of them have in common the negative impact they have on the quality of life of patients [3,8,9,10]. Patient age, body mass index, and female gender were found to be risk factors associated with the occurrence of long COVID [11,12,13].

There are several mechanisms that contribute to the pathogenesis of COVID-19 that have also been implicated in the development of long COVID. It was observed that after acute infection, some patients experience the persistence of SARS-CoV-2 in various organs, leading to chronic stimulation of the adaptive immune system and subsequent generation of cellular damage due to chronic hyperinflammation. In addition, this hyperinflammatory state can trigger hemostatic changes, such as coagulopathies [5,14,15,16]. Although some studies suggest that the inflammatory response plays an important role in the development of long COVID, information varies depending on the type of biomarker and its timeline of measurement, for example, if it is measured during acute infection or months/years after infection [3,6,17]. In addition, a possible association between the severity of acute COVID-19 infection and the development of long COVID has been observed in other studies, although the results are inconsistent. Most studies focused on taking laboratory samples when patients already have long COVID, and few studies focused on baseline laboratory parameters during the acute phase of infection like lymphocyte subpopulation, viral load, blood count, D-dimer, lactate dehydrogenase, interleukin 6, ferritin, and C-reactive protein. It was observed that patients with long COVID show elevated levels of inflammatory parameters and chemotactic and angiogenic cytokines in contrast to those patients who do not experience persistent symptoms. However, the clinical significance of these parameters has not yet been conclusively established [6,14]. While there is information available regarding viral load during the acute stage of infection and its relation to what was previously referred to as post-acute COVID-19 syndrome, various studies have yet to reach a consensus on a definitive outcome; however, they point toward a relationship between viral load and long COVID [18,19,20]. A research study conducted by Giron-Perez et al. proposed a positive correlation between viral load and the number of symptoms experienced during long COVID, implying that a higher viral load might result in a lower probability of experiencing long-term COVID symptoms [21]. A study conducted also during the early stages of the pandemic asserted that viral load can serve as a predictor for long COVID [22].

There is little information on the interrelationship between inflammatory parameters, such as C-reactive protein, lactate dehydrogenase, leukocyte count, lymphocyte count, procalcitonin, ferritin, D-dimer, interleukin-6 (IL-6), viral load level, and lymphocyte subpopulation, in the development of long COVID on a long-term follow-up. Therefore, an analysis of these parameters during the acute infection phase in hospitalized patients with COVID-19 was carried out in order to identify possible relationships between these parameters and the subsequent development of long COVID.

## 2. Materials and Methods

### 2.1. Study Population and Study Design

This study was carried out in May 2023 on patients who were hospitalized from April 2021 to January 2022 with a diagnosis of COVID-19 at Hospital Clinica Nova, a private hospital in northern Mexico. A longitudinal retrospective study was conducted following the STROBE reporting guidelines [23]. The study protocol was reviewed and approved by the Research Committee of the University of Monterrey, with registration number 25052023-CN-ENM3-CI. Since it was a retrolective study, it was not necessary to obtain informed consent from the participants.

The inclusion criteria were patients with COVID-19 confirmed with nasopharyngeal PCR who were hospitalized for severe symptoms involving decreased oxygen saturation (<94%), respiratory rate over 30 breaths/minute, and lung infiltrates > 50%. Patients from both genders, adults (18+), and those who, subsequently to the acute disease, attended COVID-19 follow-up appointments with an internist, were also included. Individuals who had previously received treatment with antivirals, steroids, convalescent plasma, or immunosuppressants prior to hospitalization as well as patients who did not have their COVID-19 variant, lymphocyte subpopulation, and viral load registered in medical history were excluded.

Various medical history data were collected, such as age, history of diabetes, systemic arterial hypertension, renal disease, chronic obstructive pulmonary disease, heart disease, among others, at the time of patient admission. Likewise, gender, oxygen requirements at admission, and vital signs were recorded.

On admission, the COVID-19 variant and lymphocyte subpopulation were measured. The time from symptoms on-set until sample collection had a median (IQR) of 8 (3) days. At admission and during hospitalization, multiple measurements were taken every 24 to 48 h including blood viral load, blood count (Sysmex XN-10, Kobe, Japan), and inflammatory parameters such as D-dimer, lactate dehydrogenase, interleukin 6, ferritin, and C-reactive protein (Roche-Cobas 6000 Module 501 & 601, Rotkreuz, Switzerland). Procalcitonin was taken on admission and in case of suspected secondary bacterial infection (DiaSorin-Liaison XL, Saluggia, Italy). Blood samples were peripheral venous punctures taken by the nursing staff. Samples regarding viral load and lymphocyte subpopulation were analyzed in the PGM Laboratory (Clinical Pathology and Genetics Laboratory), an external laboratory, which took one hour to arrive and were processed in the following two hours. The rest of the markers were analyzed in the hospital’s own laboratory during the first hour after sample collection. Results were available in the medical record within the following eight hours.

Each patient diagnosed with COVID-19 was assigned an internal medicine doctor in charge of post-disease follow-up. Depending on the severity of symptoms, patients were followed each week, every two weeks, or every month until resolution. We defined long COVID as the ongoing, relapsing, or new symptoms or conditions present 30 or more days after infection [24]. The presence and duration of long COVID symptoms were recorded in the medical history and reassessed 1.5 years after the acute infection.

### 2.2. Sample Processing Method

For the analysis of viral load, each patient had a peripheral venous blood sample collected using a tube containing a stabilizer for circulating nucleic acids (PAXGENE^®^, Mexico City, Mexico). These samples were then transported at room temperature to an external laboratory, PGM Laboratory (Clinical Pathology and Genetics Laboratory), taking on average 1 h to arrive at the PGM Laboratory from the samples’ collection [25]. At the laboratory, the samples were processed using a circulating nucleic acid extraction kit (QIAGEN^®^ Mexico City, Mexico) designed for liquid biopsy, along with a TaqPath^®^ COVID-19 kit (ThermoFisher Scientific^®^, Waltham, MA, USA) [19]. The extraction and amplification were performed using QuantStudio 5 thermal cyclers (Applied Biosystems^®^ Waltham, MA, USA). The results were subsequently transmitted to our hospital facility and uploaded into the laboratory computer system. The minimum detectable concentration of the assay was 10 copies/mL, while the maximum was 100,000 copies/mL. Based on these findings and previous reports in the literature, the following reference intervals were established for plasma results: low (<100 copies/mL), moderate (>100 to 1000 copies/mL), and high (>1000 copies/mL) [20].

For the analysis of SARS-CoV-2 variants, the samples underwent a series of procedures. Firstly, nucleic acid extraction is performed on the samples. Subsequently, retrotranscription took place, followed by a PCR reaction using a ThermoFisher Veriti endpoint thermal cycler.

The lymphocyte subpopulation was assessed using flow cytometry (BD FACS CANTO II IVD, Becton Dickinson, East Rutherford, NJ, USA). This technique allows for the extraction of lymphocytes and the analysis of various subpopulations. The parameters examined included leukocyte count, total lymphocytes, T lymphocytes (CD4 and CD8), B lymphocytes (CD19), NK cells (CD16 and CD56), and the CD4/CD8 ratio. Becton Dickinson brand antibodies were utilized in the analysis, specifically: PerCP-Cy5.5 anti-human CD45, FITC anti-human CD3, PE-Cy7 anti-human CD4, APC Cy7 anti-human CD8, APC anti-human CD19, PE anti-human CD16, and PE anti-human CD56.

### 2.3. Statistical Analysis

The distribution of the variables was assessed using the Shapiro–Wilk test and Kolmogorov test, and appropriate transformations were applied to achieve normalization when necessary. A descriptive analysis of the variables and covariates was conducted using parametric statistics, presenting means and standard deviations for variables conforming to normality, or medians and interquartile ranges for variables deviating from normality. Qualitative variables are explored using frequencies. For the continuous quantitative variables, the unpaired samples *t*-test was used to analyze variables with normal distribution, while the Mann–Whitney test was used for variables with non-normal distribution. The chi-square test was used to compare long COVID patients and non-long COVID patients. If fewer than 5 patients were in the group, Fischer’s exact test was used for univariate analysis. In addition, for a more robust model, a binary logistic regression analysis was performed to determine the association between long COVID symptoms and gender, age, inflammatory markers, ICU, and systematic arterial hypertension. A complete case analysis was conducted for missing values assumed to be missing completely at random. A value of *p* < 0.05 was considered significant. Statistical data were analyzed with SPSS vs. 25 and R v.4.0.3.

## 3. Results

Out of the cohort of 105 admitted patients with comprehensive laboratory results, a subset of 79 individuals qualified for inclusion in this study due to their adherence to follow-up appointments with the Internal Medicine Department. All had pneumonia during the acute phase. The follow-up lasted a period with a median (IQR) of 648 (68) days after the initial acute COVID-19 infection. Among the selected group of 79 participants, the median (IQR) age was 49 (22) years. A proportion of 33 individuals (41.8%) exhibited the presence of long COVID. In the long COVID group, it was observed that 22 (66.7%) were males. On the other hand, in the group without long COVID, it was found that 34 (73.9%) were males. The chi-square analysis was not significant for gender. Notably, fatigue emerged as the most recurrent symptom, shown in 15 (45.5%) patients, followed closely by tiredness, reported by another 15 (45.5%) patients. Moreover, difficulty breathing was documented among eight patients (24.2%). The patients’ medical charts did not have any psychological disturbances, even though the physician asked about them. The finer specifics of additional symptomatology data are listed in Table 1.

We observed that there was no significant difference between the variants of COVID-19 and the presence of long COVID using the chi-square test (*p* = 0.631). The variants in patients with long COVID were Delta in 19 (59.4%) patients, Alpha in 5 (15.6%) patients, Omicron in 4 (12.5%) patients, Gamma in 2 (6.1%) patients, and Beta and Epsilon in 1 (3%) patient each.

Regarding medical history, 7 (21.2%) patients with long COVID were vaccinated, while in patients without long COVID, 19 (42.2%) were vaccinated. Similarly, a chi-square analysis was performed, but no significant differences were found. The most common condition in both groups was obesity, with 23 patients (69.7%) having long COVID and 27 patients (58.7%) not having long COVID. However, no significant differences were found between the two groups in terms of obesity based on the chi-square analysis (Table 2).

Regarding respiratory treatment received in the hospital, it was observed that out of the patients with long COVID, four (12.5%) were admitted to the intensive care unit (ICU), while among the patients without long COVID, only two (4.3%) had a history of ICU admission. This difference was not statistically significant according to Fisher’s exact test. The respiratory treatment required by the patients was also compared, and the most common method was the use of nasal cannulas. Among the patients with long COVID, 26 (78.8%) required nasal cannulas, while 28 (60.9%) of the patients without long COVID used nasal cannulas. The difference in the use of nasal cannulas was not statistically significant based on the chi-square test. The remaining variables concerning respiratory support and severity disease are listed in Table 3.

The maximum peak of viral load presentation had a mean (SD) of 10 (2.64) days after the presentation of symptoms and had a non-significant median (IQR) between the long COVID and non-long COVID groups (462 (1155.82) vs. 259.0 (707.42), *p* = 0.067). Concerning the analyzed inflammatory markers, it was found that patients who developed long COVID had a higher peak of maximum lymphocytes mean (SD) during their hospitalization compared with patients without long COVID (2419.24 (1080.72) vs. 1967.15 (574.93), respectively), which was significant according to the unpaired *t*-test (*p* = 0.034). The peak of maximum leukocytes during hospitalization did not show a significant difference in median (IQR) between groups (8770 (3630) vs. 7655 (3700), *p* = 0.173). The peak of maximum lactate dehydrogenase during hospitalization did not show a significant difference in median (IQR) between groups (447.2 (224.7) vs. 399 (174.25), *p* = 0.280). The peak of maximum IL-6 during hospitalization did not show a significant difference in median (IQR) between groups (98 (434.2) vs. 89.15 (109.95), *p* = 0.846). The peak of maximum C-reactive protein during hospitalization did not show a significant difference in median (IQR) between groups (14.36 (11.84) vs. 13.79 (10.27), *p* = 0.846). The remaining inflammatory parameters during the hospital stay are listed in Table 4.

Regarding the analyzed lymphocyte subpopulation, individuals who subsequently experienced long COVID displayed an elevated count of total lymphocytes at the time of admission (1510.74 (1304.57) vs. 1133.77 (483.08), *p* = 0.025). Likewise, NK cells (CD16 and CD56) exhibited a higher presence among long COVID affected patients (224.79 (186.17) vs. 156.64 (129.72), *p* = 0.027). A comprehensive breakdown of the remaining lymphocyte subpopulations observed upon admission is listed in Table 5.

The outcomes regarding the logistic regression analysis of the risk factors and the presence of long COVID identified a positive correlation with increased levels of specific NK cell subpopulations (CD16 and CD56) (odds ratio (OR) = 1.006, *p* = 0.009). Notably, a significant positive link was also identified between the occurrence of long COVID and a prior history of ICU admission during hospitalization for acute COVID-19 (OR = 7.649, *p* = 0.045). For comprehensive further information, refer to Table 6, which provides a detailed breakdown of these findings and the non-significant variables. 

## 4. Discussion

In this study, we examined the association between long COVID 1.5 years after acute infection and various parameters recorded at the time of a patient’s hospitalization. These included inflammatory markers, clinical parameters, viral load, and lymphocyte subpopulation. Among the 79 patients studied, 41.8% of them reported experiencing long COVID symptoms, with fatigue being the most common symptom in 45.5% of cases. These findings are consistent with results from other studies reporting a long COVID prevalence of approximately 40%, although some studies reported rates as high as 85% [13,26,27]. The frequency of fatigue is approximately 40% [26]. Dyspnea has also been reported at a frequency of approximately 20%, which aligns with our results [5]. In contrast to other studies, the frequency of other symptoms such as headache and sleep disorders was not as high [4].

In the literature, it is described that the main risk factors for presenting long COVID are being female, a history of hypertension, obesity, having a psychiatric condition, and being immunosuppressed, while age was reported not to be totally related with the presence of long COVID [2,28,29]. Regarding gender, our population did not show significant differences, which could be due the sample being limited to hospitalized patients, who were predominantly male. We did not find an association with hypertension or obesity, probably due to sample size. Also, psychiatric conditions and immunosuppressed patients were not studied since there were no patients presenting these conditions. In accordance with previous studies, we did not find any association with age.

Regarding severity, our regression model revealed that admission to the intensive care unit (ICU) was a significant risk factor associated with the onset of long COVID. This finding is consistent with results from other studies, where patients who required ICU care for COVID-19 treatment exhibited subsequent symptoms encompassing physical, mental, and cognitive aspects. Notably, a study conducted by Heesakkers et al. identified weakened physical condition as the most prevalent outcome, which aligns with our investigation where chronic fatigue emerged as the predominant symptom [30,31].

Although previous studies have indicated that the amount of SARS-CoV-2 viral load correlates with the presence of long COVID and the extent of its symptoms, the current study did not find this correlation [21,22]. What sets this study apart from its predecessors is its exclusive focus on hospitalized patients, its assessment of long-term COVID symptoms extending beyond three months post-illness, and its measurement of viral load using blood samples instead of nasal swabs, which were used in previous studies. Furthermore, the current study included a smaller cohort.

It was observed that lymphocytes and specific subsets of lymphocytes, such as CD4+T cells, CD8+T cells, and natural killer cells, play an important role in maintaining the immune system function. During the COVID-19 pandemic, there has been increasing recognition of the important role that lymphocytes and their subsets play in both the clinical characteristics and treatment efficacy of the disease [32,33]. Patients with severe COVID-19 have shown a significant reduction in levels of lymphocytes, monocytes, CD4+T cells, CD8+T cells, CD3 cells, CD19 cells, and natural killer cells. These alterations were also observed in patients with long COVID [34]. Contrary to this information, our patients did not present with lymphopenia during their hospital stay. One of the main rolls of natural killer cells is to provide early defense against viral infections. In this study, we found that the patients who developed long COVID had higher levels of NK cells during their hospitalization. This could be due to the persistent immune response required to control the viral infection. However, over time, these persistent immune responses may lead to dysfunction and exhaustion of NK cells. This depletion of NK cells, along with other lymphocyte subsets, may contribute to the development and persistence of long COVID symptoms [35]. Although there is limited research on the specific role of lymphocytes and NK cells in long COVID, several studies have shown that these cells are substantially depleted in patients with long COVID [32]. The pathophysiology of long COVID is a subject that continues to be studied because it has been shown to involve not only inflammatory response influences but also many other factors. The potential increase in NK cells during acute infection followed by a decrease during long COVID could be a sign of the known redistribution and sequestration during acute infection, which leads to an increased number of NK cells in the lungs and lower levels in the blood. These sequestered cells are known to have impaired expansion and cross-talk with other immune cells [36]. This suggests the presence of immune dysregulation and probable persistent viral replication, and further explains the potential long-term effects on lymphocyte dysfunction [37,38,39].

In the multivariate model that was performed to analyze age, gender, NK lymphocyte subpopulation, ICU and arterial hypertension, ICU and the amount of NK lymphocyte subpopulation at admission were associated with long COVID. These results are similar to those obtained in other studies. For example, in a study of 89 patients, it was also found that patients who were in the ICU during acute COVID infection had a higher risk of presenting long COVID when analyzed individually. The ICU environment is associated with a higher viral load, prolonged hospital stay, and increased exposure to inflammatory biomarkers that can further contribute to immune dysregulation and depletion of lymphocytes, including NK cells [40,41]. Regarding the viral load, in our population, it was not a significant factor in the subsequent development of long COVID. In other research studies, there is little information on viral load. Some studies found that there is an association, but in other studies, no such relationship with viral load was found. The relationship between long-COVID and COVID-19 viral load is a topic of ongoing research and debate [21,41,42].

One of the main limitations of the present study is that the laboratory samples were taken only during the hospitalization of the patients. No laboratory studies were performed after that. Therefore, future research should take additional samples so that it will be possible to compare the laboratory results during the hospitalization period with laboratory results during long COVID in order to observe changes over time in lymphocytes and inflammatory markers. Also, another important aspect to consider when interpreting the results is the sample size and gender. This study was conducted in a hospitalization context, and most of the subjects were males. Therefore, the sample does not represent the whole population with long COVID. In larger investigations, more COVID-19 symptoms can be included to perform a cluster study to determine if a group of symptoms is related to a more specific parameter.

The strengths of this study are that we included patients with very complete lab tests during their hospital stay and a very long-term follow of long COVID.

## 5. Conclusions

This study showed the interrelationship between inflammatory markers, such as NK cells, and the peak of lymphocytes during acute infection of COVID-19 and the presence of long COVID during a long-term follow-up of 1.5 years in hospitalized patients. The severity of the disease in our study evaluated through admittance to the ICU also was related to the presence of long COVID.

## Figures and Tables

**Table 1 microorganisms-11-02241-t001:** Long COVID symptoms.

Variable *n* = 79	Frequency (%)
Long COVID	33 (41.8)
Fatigue	15 (45.5)
Tiredness	14 (42.4)
Difficulty breathing	8 (24.2)
Paresthesia	3 (9.1)
Palpitations	2 (6.1)
Cough	2 (6.1)
Muscle pain	2 (6.1)
Chest pain	2 (6.1)
Dysgeusia	1 (3)
Difficulty swallowing	1 (3)
Alopecia	1 (3)
Insomnia	1 (3)

**Table 2 microorganisms-11-02241-t002:** Personal history of patients.

Variable	Long COVID *n* = 33	No Long COVID *n* = 39	*p*-Value
Age	45.67 (15.95) ^a^	52.43 (17.33) ^a^	0.081 ^a^
Males	22 (66.7)	34 (73.9)	0.484
Vaccination	2 (21.2)	19 (42.2)	0.128
Obesity	23 (69.7)	27 (58.7)	0.317
Diabetes mellitus type 2	10 (30.3)	15 (32.6)	0.828
Systematic arterial hypertension	6 (18.2)	17 (37)	0.070
Asthma	5 (15.2)	2 (4.3)	0.096
Smoking	4 (8.7)	4 (12.1)	0.619
Ischemic heart failure	3 (9.1)	4 (8.7)	0.951

^a^ Data are presented as mean and standard deviation. The unpaired *t*-test was used for comparison. The remaining data are presented as frequencies and percentages. The chi-square test was used for comparison. A *p*-value < 0.05 was considered statistically significant.

**Table 3 microorganisms-11-02241-t003:** Respiratory support and severity of disease.

Variable	Long COVID *n* = 33	No Long COVID *n* = 46	*p*-Value
ICU	4 (12.5) *	2 (4.3)	0.184 ^a^
Low flow oxygenation	26 (78.8)	28 (60.9)	0.091 ^b^
High flow oxygenation	10 (30.3)	8 (17.4)	0.177 ^b^
Mechanical ventilation	4 (12.1)	2 (4.3)	0.198 ^a^
Reservoir mask	1 (3)	4 (8.7)	0.394 ^a^
Tracheostomy	2 (6.1)	0 (0)	0.910 ^a^

Data are presented as frequencies and percentages. * There was a total number of 32 patients for the ICU variable. ^a^ Fischer’s exact test was used for the comparison. ^b^ The chi-square test was used for the comparison.

**Table 4 microorganisms-11-02241-t004:** Inflammatory parameters during the hospital stay.

Variable	Long COVID	No Long COVID	*p*-Value
Hemoglobin A1c (*n* = 62)	6.005 (0.6) ^b^	6.2 (1.53) ^b^	0.701
Max. peak leukocytes (*n* = 79)	8770 (3630) ^b^	7655 (3700) ^b^	0.173
Max. peak lymphocytes (*n* = 79)	2419.24 (1080.72) ^a^	1967.15 (574.93) ^a^	0.034
Max. peak neutrophils (*n* = 79)	6920 (3170) ^b^	5550 (3773) ^b^	0.340
Max. peak lactate dehydrogenase (*n* = 75)	447.2 (224.7) ^b^	399 (174.25) ^b^	0.280
Max. peak IL-6 (*n* = 79)	98 (434.2) ^b^	89.15 (109.95) ^b^	0.846
Max. peak C-reactive protein (*n* = 79)	14.36 (11.84) ^b^	13.79 (10.27) ^b^	0.846
Max. peak D-dimer (*n* = 79)	680 (1090) ^b^	595 (455) ^b^	0.178
Max. peak ferritin (*n* = 79)	1558 (2390.5) ^b^	1468.53 (1732) ^b^	0.811
Max. peak procalcitonin (*n* = 70)	0.11 (0.2) ^b^	0.14 (0.36) ^b^	0.775
Max. peak viral load (*n* = 79)	462 (1155.82) ^b^	259.0 (707.42) ^b^	0.067

^a^ Data are presented as mean and standard deviation. The unpaired *t*-test was used for the comparison. ^b^ Data are presented as median and interquartile ranges. The Mann–Whitney U test was used for the comparison.

**Table 5 microorganisms-11-02241-t005:** Lymphocyte subpopulation at admission.

Variable	Long COVID (*n* = 33)	No Long COVID (*n* = 46)	*p*-Value
Total leukocyte subpopulation	7084.77 (2884.83) ^a^	6067.31 (2360.05) ^a^	0.089
Total lymphocyte subpopulation	1510.74 (1304.57) ^a^	1133.77 (483.08) ^a^	0.025
CD3+ T lymphocytes subpopulation	859.81 (882.7) ^b^	685.7 (550.16) ^b^	0.128
Subpopulation of helper T lymphocytes	498.53 (524.89) ^b^	408.11 (333.53) ^b^	0.223
CD8+ suppressor T lymphocyte subpopulation	255.62 (277.26) ^b^	263.74 (277.96) ^b^	0.382
B lymphocyte subpopulation CD19	164.90 (200.26) ^b^	149.96 (115.53) ^b^	0.551
Subpopulation of NK cells (CD16 and CD56)	224.79 (186.17) ^b^	156.64 (129.72) ^b^	0.027
Subpopulations CD4/CD8 ratio	2.34 (1.89) ^b^	1.81 (1.18) ^b^	0.937

^a^ Data are presented as mean and standard deviation. The unpaired *t*-test was used for the comparison. ^b^ Data are presented as median and interquartile ranges. The Mann–Whitney U test was used for the comparison.

**Table 6 microorganisms-11-02241-t006:** Binary logistic regression for prediction of long COVID.

Variable	β	Std Error	OR	*p*-Value	95%Cl
Constant	−0.243	1.090	0.784	0.823	
Age	−0.016	0.021	0.984	0.450	0.945–0.025
Gender	−0.788	0.601	0.455	0.190	0.140–1.477
Subpopulation of NK cells (CD16 and CD56)	0.006	0.002	1.006	0.009	1.002–1.011
ICU	2.035	1.015	7.649	0.045	1.045–55.972
Systematic arterial hypertension	−1.075	0.753	2.042	0.153	0.078–1.491

Adjusted R-squared = 0.273. Dependent variable: long COVID symptoms. CI, confidence interval. Std, standard.

## Data Availability

The database used and analyzed in this study is available from the corresponding author upon reasonable request.

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
