# Peer review of "Analyzing the Interplay between COVID-19 Viral Load, Inflammatory Markers, and Lymphocyte Subpopulations on the Development of Long COVID"

_microorganisms, 2023, doi:10.3390/microorganisms11092241_

Round 1
Reviewer 1 Report
The manuscript titled "Analyzing the Interplay between COVID Viral Load, Inflammatory Markers, and Lymphocyte Subpopulations on the Development of Long COVID," authored by Rivera-Cavazos et al., seeks to identify clinical and immune cell attributes that could potentially serve as predictors for individuals who develop long COVID. This analysis is based on the clinical data collected during their hospitalization.
While this work may hold interest, it lacks novelty given the numerous articles already published about inflammation and long COVID-19 and has several major issues:
1- Although the central question is intriguing, the paper predominantly utilizes previously described clinical data. Nevertheless, here factors such as age, gender, COVID-19 severity, and medical history showed no significant influence on Long-COVID development in this study. This contradicts the findings of most published papers. The authors failed to adequately address these inconsistencies, which arguably should constitute a key focus of the discussion. The limited number of individuals included in the work could be the reason for such discrepancy.
2- In attempting to unravel these disparities, it's crucial to note that certain critical information is missing. Notably, the time span between symptom onset and sample collection remains unspecified. The absence of a clear timeline for sample collection is a substantial concern. This is significant as biomarker levels and clinical data can undergo substantial changes within a mere 24 hours of hospital admission. One could postulate that differences in biomarker quantities and immune subsets might indeed impact the outcomes presented here.
3- A matter of importance is how the authors address the diverse stages at which patients enter the hospital. Extreme scenarios might involve individuals admitted with advanced respiratory distress or exhibiting only mild symptoms. These individuals might be at different disease stages upon seeking medical attention at the moment of hospitalization. Patients’ data on the symptoms developed during hospitalization should be presented.
4- Several sentences lack clarity or appear to contradict established published data. For instance, the statement, "Although some studies suggest that the inflammatory response plays an important role in the development of Long-COVID, the information is inconsistent and varies considerably," is rather surprising given the extensive body of work demonstrating the clear influence of inflammation on the course and post-COVID status of individuals.
5- Regarding the sentence, "However, the impact of Intensive Care Unit (ICU) admission as a risk factor for the development of psychological symptoms during the Long-COVID period has not yet been clearly established," its introduction seems confounding as the authors do not deal with such individuals in their study.
Some sentences are not entirely clear and need to be rephrased, like lines 90-93 on page 3. The phrase 'Individuals who had previously received treatment with antivirals, steroids, convalescent plasma, or immunosuppressants, as well as patients who did not have the COVID-19 variant, lymphocyte subpopulation, and viral load, were excluded.' What is meant by 'patients who did not have COVID-19 variant, lymphocyte subpopulation'?"
Reviewer 2 Report
Comments and suggestions
Summary section:
1. Specify the number of control visits
2. In keywords use MeSH terms
Introduction Section:
3. Mention the most relevant inflammatory markers related to your work
4. Mention more details of the relationship between viral load and long-term COVID-19
Methodology Section:
5. As an inclusion criterion, the authors indicate hospitalized patients with COVID-19. According to this, to be hospitalized they must have met the definitions of moderate or severe COVID-19. Please indicate this in methodology. What criteria must a patient have to be hospitalized for COVID-19.
Indicate whether the subsequent trace was performed only once or multiple control views were scheduled.
6. Indicate the bibliographic reference that indicates that the blood samples obtained must be transported at room temperature for the corresponding analysis.
7. Indicate the average time that the samples took from the sample taking place to the laboratory where it was processed
8. Was a validated instrument used to identify and quantify the symptoms of long-term COVID-19 or was only the clinical record of the symptoms manifested by the patient at the control visit carried out?
9. Why was a control group not included?
Results section:
10. It is mentioned that it was analyzed by variants of COVID-19, this must be explained in methodology.
Conclusion section:
11. Restructure the conclusion so that it is more concise and appropriate to the objectives of the study, not simply repeat the results.
Minor editing of English language required
Reviewer 3 Report
Analyzing the Interplay between COVID Viral Load, Inflammatory Markers, and Lymphocyte Subpopulations on the Development of Long COVID
This is a case-control retrospective single-center study aims to explore the association between Long-COVID and inflammatory markers, viral load, and lymphocyte subpopulation during acute infection in hospitalized patients. It enrolls 79 patients hospitalized for COVID-19 from April 2021 to January 2022, with a mean follow-up time of 1.5 year with the internal medicine department. The purpose of the study is interesting. The manuscript is well written and supported by literature data. However, some data are missing, and some parts needs to be explained in a more methodic way.
My concerns are detailed below:
Abstract:
The abstract adequately summarizes the content of the article. I suggest making it shorter and underling the knowledge gap and the innovation of the study: try to find early markers of long-covid development.
Introduction:
The introduction is appropriate, but too long. I would make it shorter summarizing the last paragraph about the inflammatory changes during Long-COVID. Moreover, I would not mention psychological factors in long-covid development, considering the ambiguous nature of his term for the topic at that time.
Materials and Methods:
Methods are sufficiently clear, but some data are missing.
- I would add the long-COVID definition you use to define the case group.
- All included patients presented 1 blood test collecting all the studied parameters every 24-48 h. What about viral load? Which tests were performed for the blood sample? Were them performed? Please explain it better.
- Line 86: Are you sure that is allowed not to sign the informed consent for retrospective study? We are used to make sign a general informed consent when patients are hospitalized for retrospective studies.
- Line 90: what do you mean with “who had previously received treatment with (…)”? you mean for the same COVID-19 episode or in the past? Don’t you use steroids and antivirals during the acute COVID-19 phase in Mexico? Try to specify it better, to make us understand the rational of your choice.
- Line 100: “viral load” you mean in nasopharyngeal swab maybe? Please clarify and adds a sentence about the study of lymphocyte subpopulations.
- Line 106: I would add the follow-up time at the end of the paragraph.
Results:
Results are presented in a disorganized way. Please try to give them a better shape.
- Line 162-163: goes in methods. Leave only “the chi-square analysis found not difference in ages between etc..”.
- Do you collect any data about preexistent autoimmune or immunocompromise conditions? If yes, I suggest adding it to the table A. Moreover, I would mention “age” in this table instead than in table 3.
- Why you have not included data about neuropsychiatric long-covid such as: mood change, insomnia, loss of memory and concentration. If you don’t collect this data, please mention it in study limitations.
- In table 2 why do you have 32 pts for ICU? If you have a missing data, please write it in the description below the table and leave the total N as 33 on the top of the column.
- All patients developed pneumonia during acute phase? Please add it in table 3 or in methods. Do you have any data about micro-pulmonary thromboembolism during acute phase or acute COVID-19 treatment? Please mention it in methods or results.
- Line 178 are methods. Were all patients in low-flow oxygen treated by nasal cannulas? I would mention “low-flow oxygenation” instead of “nasal cannulas”.
- Line 188: is method. Write the sentence as “concerning the analyzed inflammatory markers.. or simply inflammatory markers are shown in table 3” and then you specify.
- Line 199 goes in methods.
- Line 211 goes in methods.
- Line 237: older ages (> 65-70 years old) are not affected that much by long covid. Change this sentence according to literature data.
- You have not mentioned the viral load in the results: please do it. Have you looked for viral shedding duration?
- I would mention as for study limitation, also the bias selection population at the origin: they were all hospitalized population, so female population could be underrepresented, so the epidemiological, clinical and laboratory data presented can be not representative of the whole long-covid population.
- I suggest to better explain the possible implications of your findings and possible pathogenetic path. Find attached this article: Di Vito C, Calcaterra F, Coianiz N, Terzoli S, Voza A, Mikulak J, Della Bella S, Mavilio D. Natural Killer Cells in SARS-CoV-2 Infection: Pathophysiology and Therapeutic Implications. Front Immunol. 2022 Jun 30;13:888248. doi: 10.3389/fimmu.2022.888248. PMID: 35844604; PMCID: PMC9279859.
Conclusions:
Lines 290-290 are redundant. At the end of the paragraph add: in hospitalized patients.
English language was good and easy to understand (I'm not mother tongue). Some minor editing should be done.
Round 2
Reviewer 1 Report
The authors implemented modifications that enhanced the clarity of the text, even though the results were not robust due to the limited number of participants.
Table 3 is not referenced in the main text.
In the abstract: "ICU admission during acute COVID-19 was also linked to the development of Long-COVID symptoms." However, in Table 3, the p-value for the association between ICU admission and Long COVID is p=0.184, indicating nonsignificance according to the Fisher test. Subsequently, in the text, it is stated: "Notably, a significant positive link was also identified between the occurrence of Long-COVID and a prior history of ICU admission during hospitalization for acute COVID-19 (OR=7.649, p=0.045)."
One of the most prominent conclusions drawn is that the severity associated with ICU stay is based on the p-value of 0.184 from Table 3 and a borderline significant p-value of 0.045 from a multivariable model. Given the borderline significance in the latter case, could we consider these results as sufficiently robust to sustain the authors' conclusion?
In the paragraph between lines 280 and 291, there are repetitions, indicating the need for a rewrite of this paragraph.
